



**Monitoring potential ionospheric changes caused by Van earthquake (Mw**
**7.2)**
**Samed INYURT[1], Selcuk PEKER [2]and Cetin MEKIK[1]**
[1]Bulent Ecevit University, Geomatics Engineering Department, Zonguldak (samed_inyurt@hotmail.com,
cmekİk@hotmail.com)
[2] General Command of Mapping, Ankara,  Turkey (selcuk-peker@hotmail.com)
**ABSTRACT**
Many scientists from different disciplines have studied earthquakes for many years. As a result
of these studies, it has been proposed that some changes take place in the ionosphere layer
before, during or after earthquakes, and the ionosphere should be monitored in earthquake
prediction studies. This study investigates the changes in the ionosphere created by the
earthquake with magnitude of Mw=7.2 in the northwest of the Lake Erçek which is located to
the north of the province of Van in Turkey on 23 October 2011 and at 1.41 pm local time (-3
UT) with the epicenter of 38.75° N, 43.36° E using the TEC values obtained by the Global
Ionosphere Models (GIM) created by IONOLAB-TEC and CODE. In order to see whether the
ionospheric changes obtained by the study in question were caused by the earthquake or not,
the ionospheric conditions were studied by utilizing indices providing information on solar and
geomagnetic activities (F10.7 cm, Kp, Dst).
One of the results of the statistical test on the TEC values obtained from the both models,
positive and negative anomalies were obtained for the times before, on the day of and after the
earthquake, and the reasons for these anomalies are discussed in detail in the last section of the
study. As the ionospheric conditions in the analyzed days were highly variable, it was thought
that the anomalies were caused by geomagnetic effects, solar activity and the earthquake.
**Keywords:** TEC, Van Earthquake, Ionosphere







**1.  INTRODUCTION**





The ionosphere is the part of the atmosphere at the altitudes of 60 km to 1,100 km where there
are ions and free electrons in considerable amounts that can reflect electromagnetic waves. It
completely covers the thermosphere, one of the main layers of the atmosphere, but also includes
some of the mesosphere and the exosphere.
The most important parameter that defines the ionosphere in space and time is the quantity of
electrons. This quantity varies under the influence of the day-night cycle, seasons, geographical
location and magnetic storms in the sun. As it is not possible to measure the quantity of electrons
in the ionosphere directly, indirect measurement and calculation methods have been developed
(Li and Parrot, 2018). Total Electron Content (TEC), which is defined as the quantity of free
electrons along a cylinder with a cross section of 1 m$^2$, is a suitable parameter to monitor the
changes in the ionosphere in space and time. All signals that contain audio and data that pass
through or get reflected from the ionosphere, which is highly irregular and difficult to model,
are affected by the structure of this layer.
Calculating Total Electron Content (TEC) is a method used directly to investigate the structure
of the ionosphere. TEC is represented by the unit of TECU, and one TECU equals to
$10^{16}\ el/m^2$ (Schaer, 1999). TEC is expressed in two ways: STEC (Slant Total Electron
Content); the free electron content calculated along the slanted line between the receiver and
the satellite, and VTEC (Vertical Total Electron Content); the free electron content calculated
along the zenith of the receiver (Langley, 2002).
TEC varies based on positional and temporal variables such as the latitude of the place, seasons,
solar activity, geomagnetic storms and earthquakes. Ionospheric altitude also differs based on
geography.
TEC, which is defined as the number of free electrons on the one square meter area on the line
followed by a radio wave, is one of the important parameters for examining the structure of the
ionosphere and the upper atmosphere. With TEC values, it is possible to examine the short and
long-term changes in the ionosphere, ionospheric irregularities and disruptive factors together
(Erol and Arıkan 2005).
Ionospheric changes are being studies in more than twenty countries today as precursors of
earthquakes. Definition of ionospheric anomalies and feasibility studies of seismo-ionospheric
precursors are still ongoing (Akhoondzadeh et al., 2018; Liu et al., 2010; He et al., 2012;
Kamogawa and Kakinami, 2013;  Heki and Enomoto, 2015; Pulinets and Davidenko, 2014;
Masci et al., 2015; Yildirim et al., 2016; He and Heki, 2017; Kelley et al., 2017;Rozhnoi et al.,
2015; Thomas et al., 2017).



## 2. METHODOLOGY

### 2.1 IONOLAB-TEC Method:

The IONOLAB-TEC method developed by the department of Electrical and Electronics Engineering of Hacettepe University is a JAVA application that uses the Regularized TEC (D-TEI) algorithm (Arikan et al. 2004 ).

In this application, they developed a method that estimates VTEC values by using all GPS signals measured at a period of time in a day. While the measurements taken from the satellites with elevations of $60^o$ or higher are used, the measurements from the satellites with elevations of $10^o\ to\ 60^o$ are weighted by a Gauss function. The data from satellites with elevations of lower than $10^o$ are not included in calculations to reduce multipath effects. In this method raw GPS data was used to determine VTEC value.

### 2.2 Global Ionosphere Model (GIM):

Global Ionospheric Maps are published in the IONEX (IONosphere map EXchange) format in a way that covers the entire world. The institutions that produce these maps in the world include CODE (Center for Orbit Determination in Europe, Switzerland), DLR (Fernerkundungstation Neustrelitz, Germany), ESOC (European Space Operations Centre, Germany), JPL (Jet Propulsion Laboratory, California), NOAA (National Oceanic and Atmospheric Administration, United States), NRCan (National Resources, Canada), ROB (Royal Observatory of Belgium, Belgium), UNB (University of New Brunswick, Canada), UPC (Polytechnic University of Catalonia, Spain), WUT (Warsaw University of Technology, Poland). In this study we used the GIM-TEC values produced by CODE in the IONEX format. In the dates they were analyzed, the temporal resolution of the TEC values was 2 hours, while their positional resolution was 2.5º by latitude and 5º by longitude. In order to calculate TEC values for a point whose latitude and longitude is known on the GIM-TEC maps created by CODE using more than 300 GNSS receivers around the world, the 4 TEC values that cover the point and the two-variable interpolation formula are given below.

$$E_{int}(\lambda_0 + p\Delta\lambda, \beta_0 + q\Delta\beta) = (1-p)(1-q)E_{0.0} + p(1-q)E_{1.0} + q(1-p)E_{0.1} + pqE_{1.1} \qquad (1)$$

p and q: $0 \leq$ p, q < 1.

$\Delta\lambda$ and $\Delta\beta$: Longitude and Latitude differences grid widths,

$\lambda_0\ and\ \beta_0$: Initial longitude and latitude values,

$E_{0.0}, E_{1.0}, E_{0.1}\ ve\ E_{1.1}$ : TEC values known in neighboring points,

$E_{int}$: TEC value to be found.



**3.  ANALYSIS TO DETERMINE EARTHQUAKE-RELATED TEC CHANGES**
In order to investigate earthquake-related TEC changes, the TEC values for the stations close
to the epicenters, HAKK, MALZ, OZAL and TVAN (TUSAGA-Aktive CORS-TR) GPS
stations were analyzed to determine TEC value using the IONOLAB-TEC and GIM-TEC
models. The correlation coefficient was obtained for the TEC values from both models between
the dates 13.10.2011 and 02.11.2011 for the stations above.

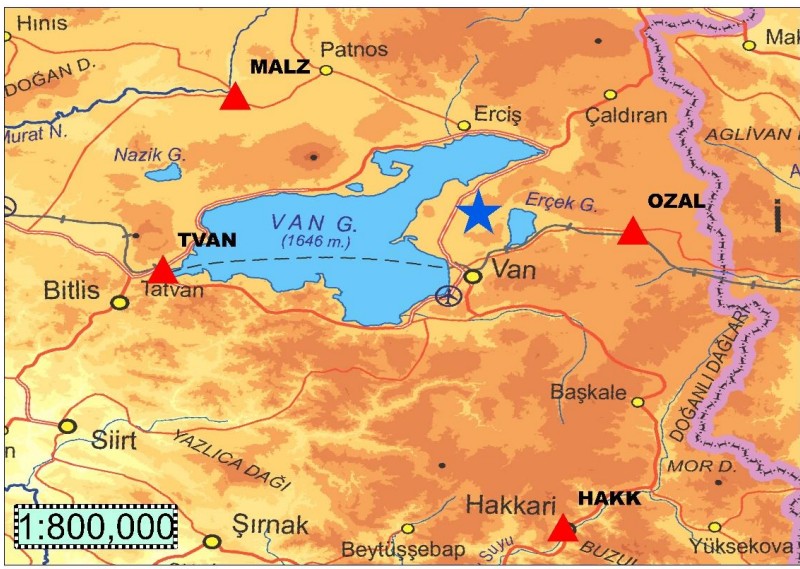

**Figure 1.** Analyzed Stations
Figure 1 shows the stations analyzed (represented by red triangles) and the epicenter of the
earthquake represented by blue star. For each station, the TEC values with the temporal
resolution of two hours obtained from both the IONOLAB-TEC and GIM-TEC models and the
correlation coefficient showing whether there is a linear relationship between two values were
calculated as below;
$$r = \frac{\sum(xy) - (\sum x)(\sum y)/n}{\sqrt{(\sum x^2 - (\sum x)^2/n)(\sum y^2 - (\sum y)^2/n)}}$$                                (2)
In order to determine the outlier values among the TEC values with a two-hour temporal
resolution from both models, the TEC values obtained from both models between the dates
01.10.2011 and 10.10.2011, which were considered quiet in terms of geomagnetic and solar
activity, were used to determine the upper boundary (UB) and the lower boundary (LB). By



utilizing the TEC values from both models, the UB and LB values were calculated using the
formulae x+3σ and x-3σ. Here, x is the mean TEC value for the relevant epoch and σ is the
standard deviation. If the TEC value in any epoch is higher than the upper boundary, it is a
positive anomaly. Similarly if it is lower than the lower boundary, it is a negative anomaly. In
order to investigate whether the anomalies before, on the day of and after the earthquake were
caused by the earthquake or not, we also examined the (Kp*10), Dst and F10.7 cm indices,
which provided information on the geomagnetic and solar activity for the days in which
anomalies were detected.

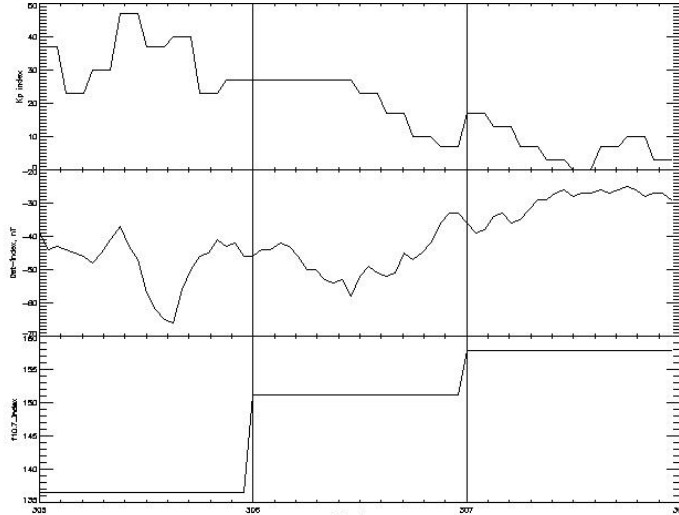

**Figure 2.** The Chart for the Dates 01-03.11.2011 in (Kp*10), Dst and F10.7 cm Indices
(URL-1)

Figures 2 shows that the (Kp*10), Dst and F10.7 cm indices that provide information on
geomagnetic and solar activity in October and on the first three days of November. Below are
the TEC values for the HAKK station for the dates 13.10.2011-02.11.2011 obtained using the
GIM-TEC and IONOLAB-TEC methods.

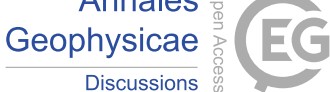



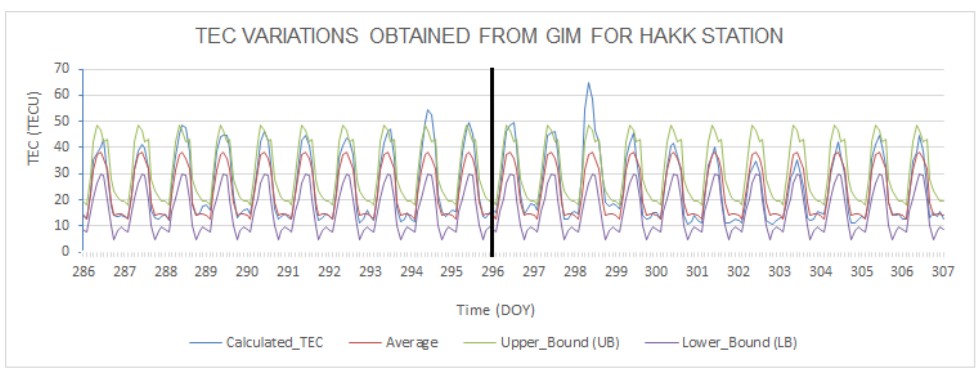


**Figure 3.** GIM-TEC Values for the HAKK Station


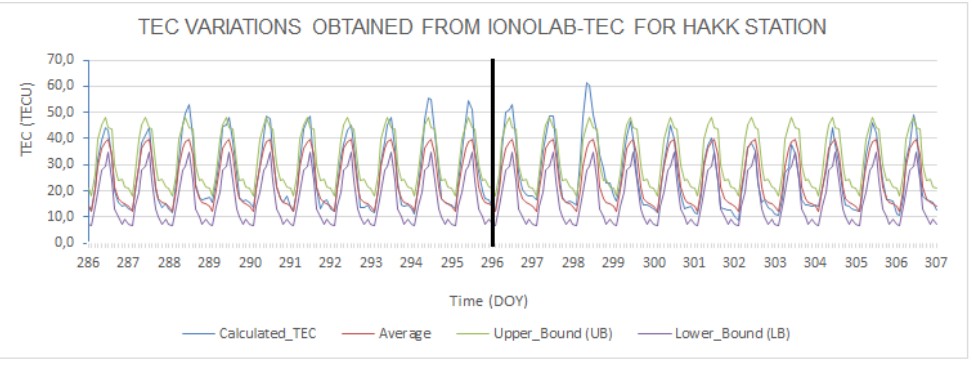


**Figure 4.** IONOLAB-TEC Values for the HAKK Station

The correlation coefficient *r* between the TEC values calculated by both methods for the HAKK
station was 0.98 indicating a strong positive relationship. The anomaly tables for this station
are provided below (Tables 1 and 2).

| GIM-TEC Anomaly Table for HAKK Station | | | | | | | | | |
|---|---|---|---|---|---|---|---|---|---|
| Number | DOY | Hour | TEC Difference (TECU) | Type of Anomaly | Number | DOY | Hour | TEC Difference (TECU) | Type of Anomaly |
| 1 | 286 | 12 | 1.0 | Positive | 7 | 294 | 12 | 10.5 | Positive |
| 2 | 288 | 12 | 5.7 | Positive | 8 | 295 | 12 | 7.3 | Positive |
| 3 | 289 | 12 | 2.5 | Positive | 9 | 296 | 12 | 7.5 | Positive |
| 4 | 290 | 12 | 0.5 | Positive | 10 | 297 | 12 | 4.1 | Positive |
| 5 | 292 | 12 | 0.8 | Positive | 11 | 298 | 8 | 16.5 | Positive |
| 6 | 293 | 12 | 5.2 | Positive | | | | | |

**Table 1.** HAKK Station Global Ionosphere Model Anomaly Table





| IONOLAB-TEC Anomaly Table for HAKK Station | | | | | | | | | |
|---|---|---|---|---|---|---|---|---|---|
| Number | DOY | Hour | TEC Difference (TECU) | Type of Anomaly | Number | DOY | Hour | TEC Difference (TECU) | Type of Anomaly |
| 1 | 287 | 12 | 0.4 | Positive | 9 | 295 | 12 | 7.2 | Positive |
| 2 | 288 | 12 | 9.2 | Positive | 10 | 296 | 12 | 8.8 | Positive |
| 3 | 289 | 12 | 4.3 | Positive | 11 | 297 | 12 | 4.6 | Positive |
| 4 | 290 | 12 | 3.8 | Positive | 12 | 298 | 8 | 16.5 | Positive |
| 5 | 291 | 12 | 4.5 | Positive | 13 | 301 | 12 | 0.3 | Negative |
| 6 | 292 | 12 | 1.4 | Positive | 14 | 302 | 14 | 0.9 | Negative |
| 7 | 293 | 12 | 4.2 | Positive | 15 | 303 | 12 | 0.7 | Negative |
| 8 | 294 | 12 | 10.9 | Positive | 16 | 306 | 10 | 0.9 | Positive |

**Table 2.** HAKK Station IONOLAB-TEC Anomaly Table

Below are the TEC values for the MALZ station obtained using the GIM-TEC and IONOLAB-
TEC methods (Figures 5 and 6).

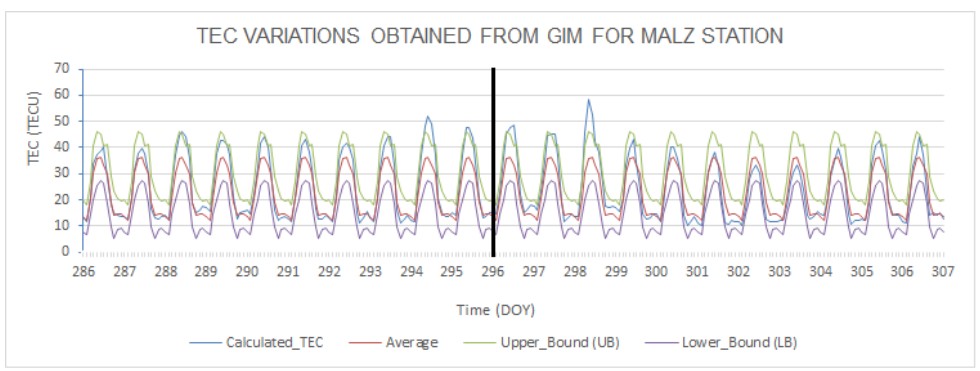


**Figure 5.** GIM-TEC Values for the MALZ Station

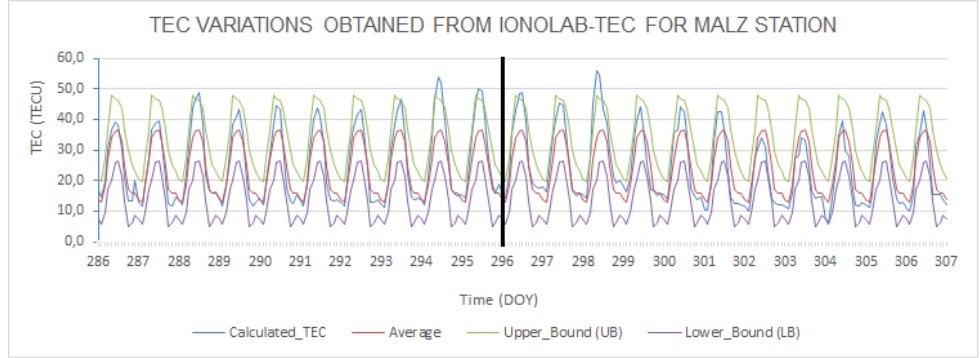


**Figure 6.** IONOLAB-TEC Values for the MALZ Station


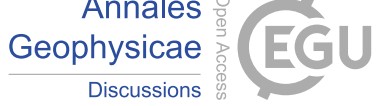


The correlation coefficient *r* between the TEC values calculated by both methods for the MALZ
station was 0.98 indicating also a strong positive relationship. The anomaly tables for this
station are provided below (Tables 3 and 4).

| GIM-TEC Anomaly Table for MALZ Station | | | | | | | | | |
|---|---|---|---|---|---|---|---|---|---|
| Number | DOY | Hour | TEC Difference (TECU) | Type of Anomaly | Number | DOY | Hour | TEC Difference (TECU) | Type of Anomaly |
| 1 | 288 | 12 | 3.5 | Positive | 5 | 295 | 12 | 3.1 | Positive |
| 2 | 289 | 12 | 0.5 | Positive | 6 | 296 | 12 | 7.9 | Positive |
| 3 | 293 | 12 | 3.9 | Positive | 7 | 297 | 12 | 4.7 | Positive |
| 4 | 294 | 12 | 8.6 | Positive | 8 | 298 | 8 | 12.6 | Positive |

**Table 3.** MALZ Station Global Ionosphere Model Anomaly Table


| IONOLAB-TEC Anomaly Table for MALZ Station | | | | | | | | | |
|---|---|---|---|---|---|---|---|---|---|
| Number | DOY | Hour | TEC Difference (TECU) | Type of Anomaly | Number | DOY | Hour | TEC Difference (TECU) | Type of Anomaly |
| 1 | 288 | 12 | 2.3 | Positive | 5 | 296 | 12 | 2.5 | Positive |
| 2 | 293 | 12 | 0.4 | Positive | 6 | 298 | 6 | 8.6 | Positive |
| 3 | 294 | 10 | 7.4 | Positive | 7 | 304 | 0 | 0.2 | Negative |
| 4 | 295 | 10 | 3.6 | Positive | | | | | |

**Table 4.** MALZ Station IONOLAB-TEC Anomaly Table
Tables 3 and 4 show the anomalies found as a result of the analysis of the TEC values obtained
by the IONOLAB-TEC and GIM-TEC methods for the MALZ station. It is believed that the
positive anomaly on days 288 and 289 was caused by moderate (136.9 sfu, 150 sfu) solar
activity. It is also believed that the anomalies on the days 293, 294, 295 and 296 were caused
by strong (157.8 sfu, 166.3 sfu, 162.5 sfu, 153.9 sfu) solar activity.

Below are the TEC values for the OZAL station obtained using the GIM-TEC and IONOLAB-
TEC methods for the dates 13 October – 02 November (Figures 7 and 8).



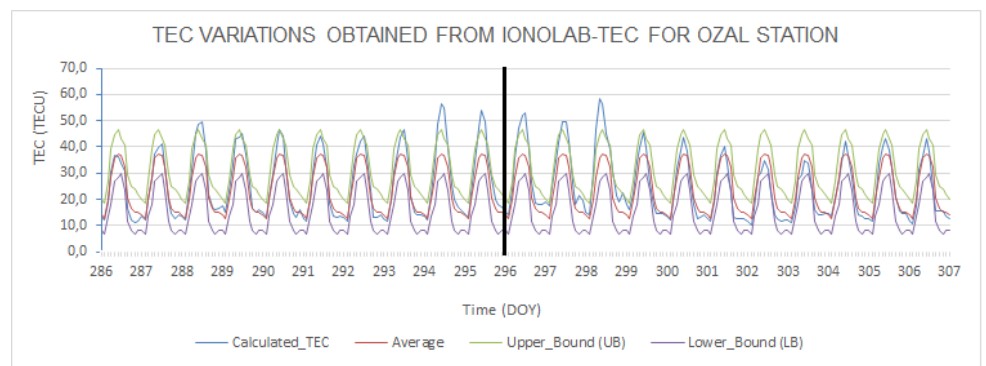


**Figure 7.** GIM-TEC Values for the OZAL Station
**Figure 8** IONOLAB-TEC Values for the OZAL Station

The correlation coefficient *r* between the TEC values calculated by both methods for the OZAL
station was 0.98 demonstrating a strong positive relationship. The anomaly tables for this
station are provided below (Tables 5 and 6).

| GIM-TEC Anomaly Table for OZAL Station | | | | | | | | | |
|---|---|---|---|---|---|---|---|---|---|
| Number | DOY | Hour | TEC Difference (TECU) | Type of Anomaly | Number | DOY | Hour | TEC Difference (TECU) | Type of Anomaly |
| 1 | 288 | 12 | 2.8 | Positive | 5 | 296 | 12 | 7.2 | Positive |
| 2 | 293 | 12 | 3.2 | Positive | 6 | 297 | 12 | 4.0 | Positive |
| 3 | 294 | 12 | 7.9 | Positive | 7 | 298 | 8 | 12.4 | Positive |
| 4 | 295 | 12 | 2.4 | Positive | | | | | |

**Table 5.** OZAL Station Global Ionosphere Model Anomaly Table




| IONOLAB-TEC Anomaly Table for OZAL Station | | | | | | | | | |
|---|---|---|---|---|---|---|---|---|---|
| Number | DOY | Hour | TEC Difference (TECU) | Type of Anomaly | Number | DOY | Hour | TEC Difference (TECU) | Type of Anomaly |
| 1 | 288 | 12 | 6.1 | Positive | 7 | 295 | 10 | 7.4 | Positive |
| 2 | 289 | 12 | 1.6 | Positive | 8 | 296 | 12 | 9.6 | Positive |
| 3 | 290 | 12 | 0.9 | Positive | 9 | 297 | 12 | 6.0 | Positive |
| 4 | 293 | 12 | 3.5 | Positive | 10 | 298 | 8 | 13.6 | Positive |
| 5 | 292 | 12 | 0.6 | Positive | 11 | 301 | 14 | 1.2 | Negative |
| 6 | 294 | 12 | 11.8 | Positive | 12 | 302 | 14 | 1.4 | Negative |

**Table 6.** OZAL Station IONOLAB-TEC Anomaly Table

Below are the TEC values for the TVAN station obtained using the GIM-TEC and IONOLAB-TEC methods (Figures 9, 10).

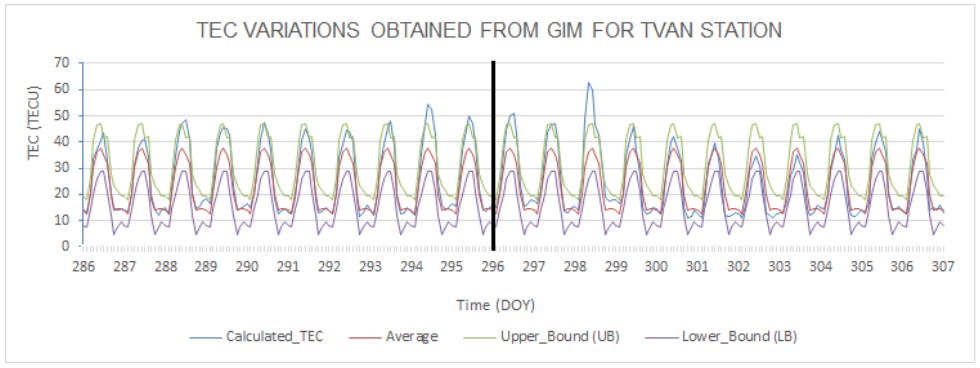

**Figure 9.** GIM-TEC Values for the TVAN Station

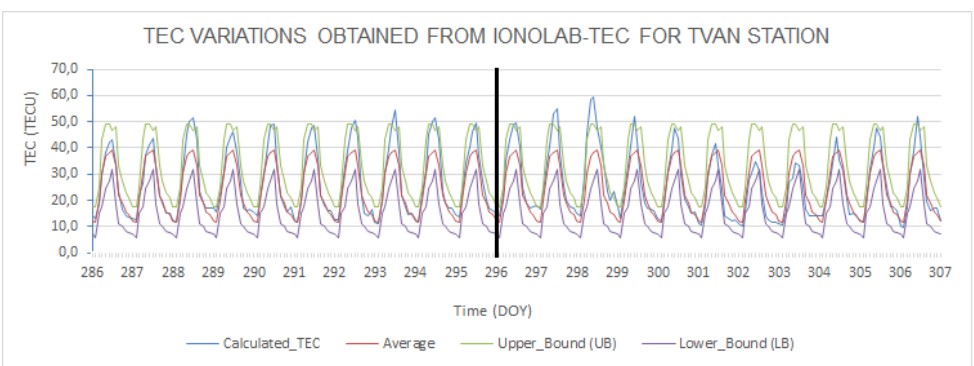

**Figure 10.** IONOLAB-TEC Values for the TVAN Station



The correlation coefficient between the TEC values calculated by both methods for the TVAN
station was 0.98 representing a strong positive relationship. The anomaly tables for this station
are provided below (Tables 7 and 8).

| GIM-TEC Anomaly Table for TVAN Station | | | | | | | | | |
|---|---|---|---|---|---|---|---|---|---|
| Number | DOY | Hour | TEC Difference (TECU) | Type of Anomaly | Number | DOY | Hour | TEC Difference (TECU) | Type of Anomaly |
| 1 | 286 | 12 | 2.1 | Positive | 10 | 294 | 12 | 11.0 | Positive |
| 2 | 288 | 12 | 7.0 | Positive | 11 | 295 | 12 | 5.4 | Positive |
| 3 | 289 | 12 | 3.5 | Positive | 12 | 296 | 12 | 9.3 | Positive |
| 4 | 290 | 12 | 1.8 | Positive | 13 | 297 | 12 | 5.5 | Positive |
| 5 | 292 | 12 | 2.8 | Positive | 14 | 298 | 8 | 16.5 | Negative |
| 6 | 293 | 12 | 6.4 | Positive | | | | | |

**Table 7.** TVAN Station Global Ionosphere Model Anomaly Table


| IONOLAB-TEC Anomaly Table for TVAN Station | | | | | | | | | |
|---|---|---|---|---|---|---|---|---|---|
| Number | DOY | Hour | TEC Difference (TECU) | Type of Anomaly | Number | DOY | Hour | TEC Difference (TECU) | Type of Anomaly |
| 1 | 288 | 12 | 5.1 | Positive | 10 | 296 | 12 | 3.4 | Positive |
| 2 | 290 | 12 | 2.6 | Positive | 11 | 297 | 12 | 8.5 | Positive |
| 3 | 291 | 12 | 2.0 | Positive | 12 | 298 | 10 | 10.5 | Positive |
| 4 | 292 | 12 | 4.0 | Positive | 13 | 299 | 10 | 2.8 | Positive |
| 5 | 293 | 12 | 8.1 | Positive | 14 | 302 | 12 | 0.7 | Negative |
| 6 | 294 | 12 | 5.1 | Positive | 15 | 306 | 10 | 2.9 | Positive |
| 7 | 295 | 12 | 3.2 | Positive | | | | | |

**Table 8.** TVAN Station IONOLAB-TEC Anomaly Table

Tables 1, 2, 3, 4, 5, 6, 7 and 8 show the results of the statistical analysis of the TEC values
created by the IONOLAB-TEC and GIM-TEC methods. The tables also depict the day and hour
in which anomalies were observed, and the quantity and type of the anomaly. The numbers of
anomalies obtained in both models were very close to each other. The F10.7 cm index values
between the days 286 and 292 were 136 sfu, 135.4 sfu, 136.9 sfu, 150 sfu, 151.6 sfu, 145.7 sfu,
146.1 sfu. The index values show that there was usually moderate solar activity. Therefore, the
anomalies in question may be related to the earthquake or solar activity. The index values for
the days 293, 294, 295 and 296 (the day of the earthquake) were 157.8 sfu, 166.3 sfu, 162.5 sfu
and 153.9 sfu respectively. These values indicate strong solar activity. On the other hand, the
ionosphere layer was quiet in these days in terms of geomagnetic conditions. As there was
strong solar activity, the numbers of anomalies were higher than the numbers in the days 286-



292. Since solar activity was moderate in the day 297, the number of anomalies dropped. The
solar activity on the day 298 was moderate, but there was strong geomagnetic activity (Dst -
147 nt, Kp*10=73). The reason for the high numbers of anomalies on day 298 in both models
is believed to be due to geomagnetic activity. Considering the analyzed days in general, it may
be seen that it is difficult to identify earthquake-related anomalies as the solar activity and
geomagnetic conditions before and after the earthquake were not quiet. Therefore, it is believed
that the anomalies detected in the stations on days 293-296 may be related to the earthquake
and/or solar activity, and the anomalies on days 297 and 298 may be related to the earthquake,
solar activity and/or geomagnetic activity.
**4. CONCLUSION**
In the scope of this study, the TEC values for the stations HAKK, MALZ, OZAL, TVAN were
obtained using the GIM-TEC and IONOLAB-TEC methods. In the comparison of the obtained
values, it was seen that there was high correlation between the TEC values obtained by the two
models. In order to detect earthquake-related TEC changes better, the TEC values created from
both models for the period of 13.10.2011-02.11.2011 were used as reference to determine the
UB and LB values. As a result of the statistical test, anomalies were found in all analyzed
stations for before, on the day of and after the earthquake. In order to understand whether the
anomalies obtained in both models were earthquake-related, the ionospheric conditions,
geomagnetic activity and solar activity on the analyzed days were examined using the Kp, Dst
and F10.7 cm indices.
Consequently, it was determined that the positive anomalies observed on days 286-292 may be
related to moderate solar activity and/or the earthquake, and the positive anomalies observed
on days 293, 294, 295, 296 (day of the earthquake) may be related to strong solar activity and/or
the earthquake. Moderate solar activity and strong geomagnetic activity were observed for day
298, so the numbers of anomalies in both models increased dramatically. This increase is
considered to be related to geomagnetic activity. The anomaly on day 298 may be related to the
earthquake, geomagnetic effects and/or solar activity. The finding that the ionospheric
conditions were variable in the analyzed days makes it highly difficult to identify earthquake-
related ionospheric changes. Therefore, interdisciplinary studies are needed to determine the
earthquake-related part of the change in question.






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
