# Peer review of "Monitoring potential ionospheric changes caused by Van earthquake (Mw"

_Annales Geophysicae, 2018_

## Referee Comment (RC1) · Anonymous Referee #1 · 15 Oct 2018

Aim of this study is to determine anomaly which caused by earth Van earthquake or the other factors like geomagnetic storms, solar activity. Van earthquake is highly complex earthquake due to including magnetic storms and solar activity. Your study comes interesting in terms of using two different models to detect ionospheric anomaly. Some anomalies were detected before and after the earthquake. Its' causes were explained detaily and results are convenient with the reality. I suggest some minor revisions to be published. Wording and misprints; Page 2 paragraph 2 line 1 throughout this paragraph replace "quantity" with "amount" or "number" or another more suitable word Page 2 last paragraph "are being studied" should be "have been studied" Page 12, conclusion, paragraph 1 line 6 for comfort for those readers who will read only abstract and conclusion write upper boundry and lower boundry instead of "UB" and "LB".

---

## Referee Comment (RC2) · Anonymous Referee #2 · 22 Oct 2018

The manuscript analyzes ionospheric changes before and after the 2011 Van earthquake using GPS TEC and GIM data. It is hard to give a comprehensive and solid support to the conclusion based on the current data analysis. I think the work should be extensively expanded for possible publication. The following suggestions might be helpful for the further improvement of this manuscript.

The main results are shown in Figures 3-10 and Tables 1-8, however, these figures and tables derived from four GPS stations and GIM TEC data are similar. I think it is enough to show the results using one typical station or combining these figures into one or two figures. The tables have the same problem.

The authors consider the solar and geomagnetic activity in this manuscript. However, the results and conclusions are not supported by the limited results. In Turkey, there are about 20 GNSS stations around the epicenter (Figure 1 in Rolland *et al*., 2013, GRL) within the studied period, therefore, it is possible to obtain the temporal and spatial distribution characteristics of the ionospheric changes. Please include the spatial analysis based on TEC data from local GPS network, at least using the GIM data.

The manuscript did not involve a discussion section, which results in an overall lacking of a proper evaluation for the results, and their connection with previous observations.

Line 41: we do have direct measurements in the ionosphere using various techniques.

Line 45: audio is a very unpractical word for this context.

Lines 43-45, 48-50, 57-59: TEC has been repeatedly defined. The introduction should be logically reorganized.

Line 95: Please add a reference for Equation 1.

Line 111: Figure 1 should add labels in x- and y-axis and the caption are too simple.

Line 118: Equation 2 is not correct.

Line 134: The time series of F10.7 in Figure 2 is not realistic, and the resolution of this figure should be improved.

Lines 148-149, 164-165, 185-186, 202-203: The correlation between the GIM and GPS TEC should be high because the CODE-GIM used IGS stations around the epicenter. It is not necessary to repeat these sentences several times.

---

## Author Comment (AC1) · 9 Nov 2018

**Monitoring potential ionospheric changes caused by Van earthquake (Mw**

2

**7.2)**

**Samed INYURT1, Selcuk PEKER 2and Cetin MEKIK1**

[revised manuscript text omitted]

88 p and q:
$$\le p, q

---

## Author Comment (AC2) · 9 Nov 2018

**RESPONSE TO REVİEWER 1**

Thank you for your valuable comments. We have edited Page 2 paragraph 2 line 1, Page 2 last paragraph, and conclusion paragraph 1 line 6 as you suggest.

---

## Author Comment (AC3) · 9 Nov 2018

**RESPONSE TO REVİEWER 2**

Thank you for your valuable comments. We have organized our paper as you suggest.

1- You suggest that figure 3-10 and table 1-10 are similar and therefore one gps station is enough to analyze. We have analyzed one gps station which is nearest receiver epicenter of earthquake.
2- You have also suggest that teporal and spatial resolution should be taken into account using GIM model. Therefore we have analyzed spatial and temporal analysis as you state
3- Discussion section was given as you suggest. Other conclusions were discussed and compared our results.
4- Line 41, line 45, line43-45, 48-50, 57-59 were defined repeatedly
5- Line 95 reference was added, line 111 x and y axis and caption was  again drawn as you state.
6- Equation 2  was edited correctly.
7- Fig 2 was drawn again as you state.
8- Line 148-149, 164-165, 185-186, 202-203 was organized again.

---

## Author Response (AR2)

**RESPONSE TO EDITOR**

Thank you for your valuable comment. We have edited our manuscript as you suggest;

1- Line 21 was edited as you said
2- Line 51 was deleted as you suggest
3- This study's aim has explained with seperate paragraph.
4- Equation 1 is referenced.
5- Line 68 was edited
6- Fig1 caption is edited
7- Equation 2 is deleted
8- İt is writen wrongly. It is edited again
9- Fig3 is edited as you suggest
10- Fig4 is edited as you suggest
11- İt was explained and referenced
12- İt was edited
13- İt was edited
14- İt was edited
15- İt was edited
16- İt was edited
17- İt was edited as you suggest
18- I explained detaily
19- He et al. 2012 study  expalined in discussion

.